# Realistic and complex visual chasing behaviors trigger the perception of intentionality

**Mohan Ji** *, **Emily J. Ward**, **C. Shawn Green**

Department of Psychology, McPherson Eye Research Institute, Madison, Wisconsin, United States of America

* mji24@wisc.edu

## Abstract

We not only perceive the physical state of the environment, but also the causal structures underlying the physical state. Determining whether an object has intentionality is a key component of this process. Among all possible intentions, the intention that has arguably been studied the most is chasing—often via a reasonably simple and stereotyped computer algorithm ("heat-seeking"). The current study investigated the perception of multiple types of chasing approaches and thus whether it is the intention of chasing that triggers the perception of chasing, whether the chasing agent and the agent being chased play equally important roles, and whether the perception of chasing requires the presence of both agents. We implemented a well-studied wolf chasing a sheep paradigm where participants viewed recordings of a disc (the wolf) chasing another disc (the sheep) among other distracting discs. We manipulated the types of chasing algorithms, the density of the distractors, the target agent in the task, and the presence of the agent being chased. We found that the participants could successfully identify the chasing agent in all conditions where both agents were present, albeit with different levels of performance (e.g., participants were best at detecting the chasing agent when the chasing agent engaged in a direct chasing strategy and were worst at detecting a human-controlled chasing agent). This work therefore extends our understanding of the types of cues that are and are not utilized by the visual system to detect the chasing intention.

## Introduction

The human visual system not only delivers information about the physical state of the environment, but it also provides information about the causal structures that underlie the physical state. This latter aspect is particularly critical for agents, such as human beings, who must act within the world, rather than simply passively observing. Indeed, an understanding of the simple physical state of the world is often insufficient to plan appropriate actions; instead, knowing the causes of the physical state is frequently required to guide proper behaviors.

For example, when viewed from a far distance, a boulder rolling down a mountain slope might have similar low-level visual properties as a bear running down the same slope (e.g., a

**Data Availability Statement:** All data, study material, and analysis files are available from the OSF database (https://osf.io/gajv2/).

**Funding:** MJ and CSG received the David G. Walsh Graduate Student Support Initiative from

McPherson Eye Research (https://vision.wisc.edu). The funders had no role in study design, data collection and analysis, decision to publish, or preparation of the manuscript.

**Competing interests:** The authors have declared that no competing interests exist.

large brown object moving toward us at some given speed). If our visual system provided only information about the physical state of the world, these two objects–boulder and bear–would therefore be treated as very similar. This would obviously be a significant error given the goal of taking an action to avoid these objects. If the object is a boulder, it is important to know that boulders are inanimate objects whose movements are thus driven exclusively by physical laws. Some initial force was applied to the boulder to start its motion after which gravity will serve to accelerate the boulder as it moves down the hill, while interactions with the ground, bushes, trees, etc. will serve to slow and/or deflect the course of the boulder. If the object, meanwhile, is a bear, it is of particular importance to know that bears are animate and intentional. In other words, the bear has internal goals that it is attempting to reach, which very might include chasing you. Thus, despite having similar low-level visual properties, once the relevant causal information is considered, it should be clear that the best course of action is very different depending on whether the object is a boulder or bear.

Because whether or not an object has intentionality is a key determinant of the types of causal inferences that should be made, it is perhaps not surprising that assessments of intentionality appear to be an important function of the human visual system [1–5]. Indeed, over the past 80+ years there has been significant interest in considering those situations where we infer that objects are or are not acting in an intentional manner. For instance, Barrett and colleagues [1] demonstrated that participants could accurately judge different types of intentional behaviors exhibited by simple geometric shapes including chasing, courting, following, etc. This, along with a great deal of other work, has together indicated that the perception that objects are being guided by any number of different intentions, including attracting, repulsing, and/or avoiding other objects, can be induced by careful design of the objects' movement patterns [6, 7].

Yet of all possible intentions, perhaps the most examined in the perceptual literature to-date is chasing (i.e., where one moving object is attempting to 'catch' another). Objects that are perceived as deliberating chasing have been shown to strongly capture attention [8, 9] and are prioritized in memory [10], thus indicating the importance of this intention. In a seminal series of studies, Gao and colleagues [11] investigated a number of aspects related to the perception of chasing utilizing a "wolf chasing a sheep" task. In the most basic version of the task, participants observed several moving discs. On some trials, all the discs moved randomly. On other trials, while most of the discs moved randomly, one disc ('the wolf') chased a second disc ('the sheep') via a predetermined algorithm. The participants' task was to determine whether a 'wolf' (i.e., chasing disc) had been present, and if so, which disc was the wolf. Participants were reasonably sensitive to this type of chasing intention and could both detect trials that contained a wolf as well as could identify the wolf itself on trials in which it was present. The authors further showed that the capacity for identifying intentional chasing could be disrupted via several different means. In particular, participants were seen to be sensitive to what the authors dubbed "chasing subtlety": participants were by far the most accurate at detecting and identifying the wolf when the wolf was controlled by a fully "heat seeking" algorithm (i.e., at each and every time point the wolf moved directly toward the sheep) and became progressively less accurate when the direct heat seeking was disrupted in some manner (e.g., if the wolf alternated between direct heat seeking and random movements, or if rather than all direct movements toward the sheep, the movements were instead drawn from a distribution centered on the direct orientation).

While the previous work that has used the wolf chasing sheep paradigm has significantly added to our understanding of the conditions under which individuals will be able to identify/detect intentional chasing, questions remain regarding whether alternative chasing behaviors will still trigger the perception of intentionality. Most previous work has used a single

computer chasing algorithm that is potentially somewhat artificial compared to what chasing events might look like in real life [11]. For instance, real-world chasing agents may not use a pure strategy approach (e.g., they may not always make direct movements toward the target) nor a mixture of pure strategy and pure noise (i.e., sometimes direct chasing, sometimes making random movements) either across trials or even within trials, as has been instantiated in previous work. Instead, real-world agents may use more complex strategies (particularly if they know that the agent they are chasing also has agency and intentionality—in this case to avoid being caught). Real-world agents may even make deliberate efforts to mask, as best as possible, their intentionality, in order to avoid detection.

Understanding the relationship between more ecologically valid but less well-controlled chasing behaviors and the more artificial but better-controlled pure strategies commonly used in laboratory studies is key for understanding the real-world applicability of the basic science approaches that have been used in the field to-date. Therefore, in Experiment 1, we aimed to test whether participants could still identify the chasing agent when it was controlled by *a human being*, and not a computer algorithm. We reasoned that while human-controlled wolves would still have intentionality (i.e., they would have the goal of catching the sheep), they might exhibit both different and more variable behaviors than those shown by the wolves controlled by a simple computer algorithm. To preface our results, we found that participants could indeed still identify the human-controlled wolves, though to a significantly lesser extent than those controlled by a simple direct-chasing computer algorithm.

Next, because in Experiment 1 we saw that human viewers were less able to identify human-controlled wolves, we considered that the intention of chasing may manifest via many different movement patterns beyond that which has been most commonly examined (i.e., direct chasing/heat seeking). If it is really the *intention* of chasing another agent that is triggering the perception of chasing, we should observe similar performance in terms of identifying the chasing agent for other types of movement algorithms that accomplish the same goal or are associated with the same intention. Thus, in Experiment 2 we contrasted participants' ability to identify a chasing agent that was controlled by one of three possible algorithms: direct chasing, sheep mimicking, and sheep interception. In this experiment we also examined two additional issues, which have also been considered in past research. The first was whether the ability to identify chasing agents was impacted by the number of distractor discs present in the display. It has been previously argued that chasing behaviors are given attentional priority [8, 9], which if so, could suggest that adding distractors would minimally impact performance (in the same way that adding distractors produces a minimal impact in terms of reaction time in efficient/pop-out search displays). The second issue was whether simple patterns of correlated movement with the sheep would be sufficient to drive the perception of intentionality. To examine this question, in all trials we included a distractor disc that "chased" the mirror reverse of the true sheep (meaning that the degree of correlation between the true wolf and sheep and "correlation-control" sheep were identical). We then assessed how often participants mistakenly chose this alternative mirror-chasing wolf. To preview our results, we found that identification accuracy was much higher than chance across several different chasing algorithms, suggesting that participants' ability to identify the chasing intention is not specific to just direct heat-seeking behaviors. However, performance was not identical across algorithms, suggesting that some types of movement behavior better fit participants' "template" for what chasing should look like. We also found that performance for all algorithms diminished with increasing numbers of distractors. And finally, we found that simple correlated movements were not sufficient to drive the perception of intentionality, as participants essentially never chose the "correlation-control" wolf (video available on https://osf.io/gajv2/).

Next, previous work has suggested that a particular relationship between the movement of the sheep and wolf is necessary to produce the perception of chasing. For instance, when the correlation between the movement of the sheep and the wolf is reduced by introducing random movements in the wolf trajectory or wolf movements that are less perfectly directed toward the sheep, the perception of chasing is also diminished [11, 12]. Yet it is notable that participants' tasks in these experiments have nearly always focused on the wolf (i.e., to detect the wolf, to identify the wolf, to avoid the wolf). We thus examined whether both roles (sheep & wolf) were equivalently easy to identify. On one hand, given that patterns of related motion between the wolf and sheep appear to be at the root of participants' ability to identify chasing, it should be the case that either role could be identified (i.e., there's no way to know that the movement patterns of two discs is related without knowing the identity of both discs). On the other hand, it is arguably more ecologically important to detect a chasing agent rather than an agent being chased, in which case the ability to detect the wolf could exceed that of the sheep. Thus, in Experiment 3, we had participants attempt to identify either the sheep or the wolf in separate trials. As we outline below, our results did not support the former point of view; participants were actually better at identifying the agent being chased (i.e., the sheep) as compared to the chasing agent (i.e., the wolf).

Lastly, we examined whether it was truly necessary for the sheep to be visible for the wolf to be identified as an intentional agent. As will be discussed in detail below, while we had generated our distractor discs in Experiment 2 in such a way to minimize the extent to which their movement patterns were grossly different from those of the distractor discs, the cleanest way to ascertain that both sheep and wolf are necessary for the perception of chasing is to simply make the sheep invisible (i.e., the wolf moved the same way as always, but there was no visible sheep for the participant to observe). As will be shown in the results of Experiment 4, this resulted in a complete inability of participants to detect the wolf, suggesting that it is indeed the interplay between wolf and sheep that produces the perception of chasing, rather than simple statistical aspects of the wolf motion alone.

## Experiment 1: Detection of a chasing agent controlled by a computer algorithm versus a human

Previous studies have established that humans are able to detect intentional behaviors such as chasing in a noisy environment. However, in these experiments, the chasing agent, as well as the agent being chased, have commonly been generated using fixed algorithms that produced rather simplistic movement patterns. It is thus unclear whether participants can detect the chasing agent when either intentional agent (i.e., the chasing agent and/or the agent being chased) exhibits more variable movement patterns, such as would be the case if the wolf and/or sheep were controlled by human participants. Here we sought to ask the question whether participants could successfully identify the chasing agent in a "wolf chasing sheep" paradigm where either the sheep alone was controlled by a human (avoiding a computer-controlled wolf) or where both the sheep and the wolf were controlled by human players. The participants' goal on each trial was to identify the wolf, and thus performance was measured in terms of identification accuracy. We expected that the human-controlled wolf trials would be associated with lower accuracy than computer-controlled wolf trials, but that identification rate will still be above chance.

### Methods

Experiment 1 consisted first of a stimulus generation phase (i.e., in order to produce videos of human and computer-controlled wolves chasing human sheep) that was then followed by the

perception task phase (i.e., where we assessed the extent to which human participants could identify the wolf in the aforementioned videos).

### Stimulus generation phase

**Participants.** A total of 10 participants took part in the stimulus generation phase after providing informed consent. All participants worked as research assistants in author CSG's lab. Experimental procedures were approved by the University of Wisconsin—Madison Institutional Review Board.

**Apparatus and display.** All stimuli were generated using MATLAB (R2019a; Math-Works, Natick, MA) using the Psychophysics Toolbox 3 [13–15]. Participants in the stimulus generation phase completed the entire task in MATLAB in person.

**Stimuli.** All stimuli were presented within a black display frame subtending 32˚ by 24˚ of visual angle (calculated using a viewing distance of 50cm, although precise seating distance was not enforced). Three different types of stimuli (small discs) were present on the screen on each trial: the sheep, the wolf, and normal distractors.

Human participants either controlled the sheep alone and played against a computer-generated wolf (with the goal of avoiding being captured by the wolf for the entire trial duration) or one human participant played as the sheep against a wolf that was also controlled by a human participant (the sheep's task was to avoid being caught, the wolf's task was to catch the sheep).

From the perspective of the human participant playing as the sheep, their disc was colored green (0.87˚ in diameter, Fig 1A). The wolf was white in color (1.3˚ in diameter) and was

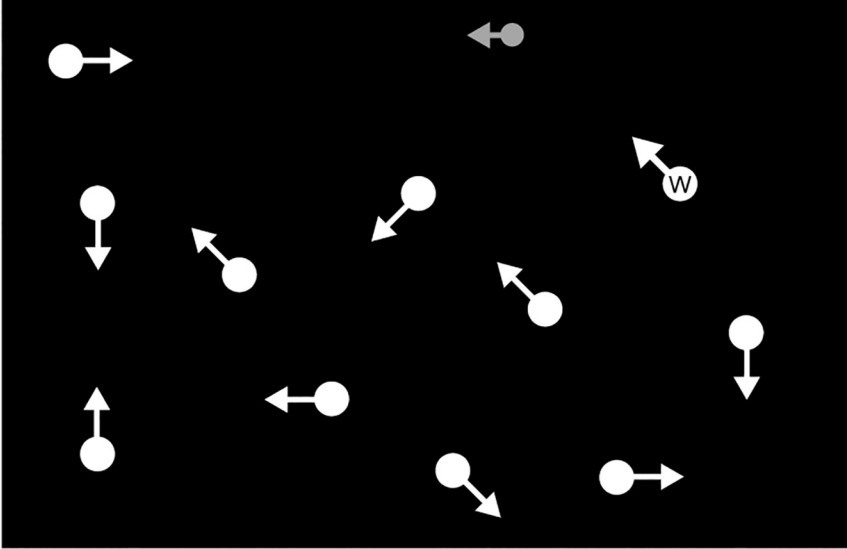
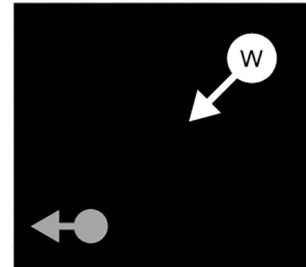
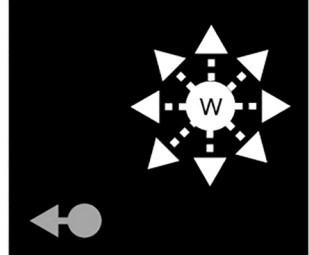

**Fig 1.** (A) Stimuli were generated in a separate stimulus generation phase where human participants controlled either the sheep-alone (gray disc) against a computer-controlled wolf (white disc marked with a "W"; note that for the human participant playing as the sheep, the wolf would have been without the letter mark, the same as the distractor discs) or else different humans controlled the sheep and wolf. Note that for clarity of illustration purposes only 10 distractors are shown, but the actual display contained 19 distractors. The "human sheep" view was then translated into movie format for use in Experiment 1. (B) Human-controlled sheep were tasked with avoiding a wolf that was controlled either by the computer, guided by a direct chasing algorithm (top panel), or else that was controlled by another human being (right panel), that could decide for herself/himself at each moment in time which direction to go in order to most effectively catch the sheep.

initialized with a minimum distance of 11˚ from the sheep. The remainder of the discs (19) were distractors (white; 1.3˚ in diameter). All distractors were randomly initialized within the display. Meanwhile, from the perspective of the human participant playing as the wolf, their disc was colored green (so the disc controlled by the participant was always the same color from their individual perspectives), the sheep was in red, and the distractors were white.

Participants used a keyboard to control the sheep and/or wolf and were only allowed to move in eight directions (up, down, left, right, and the four ordinal directions). The sheep moved at a speed of 14.4˚/s. The wolf and distractors moved at a slightly slower speed of 9.6˚/s (thus together, the size, color, and speed of the wolf and distractors were matched meaning that none of these low-level features could be used to infer the identity of the wolf).

The computer-controlled wolf's movement was controlled by a direct chasing algorithm (Fig 1B). Here, the angle between the sheep and the wolf was calculated at each time point and the wolf moved in the allowable direction closest to that angle (i.e., as close as "directly at the sheep" as possible, given that it could move in only 8 distinct directions). The distractors always moved in a random fashion while changing their direction every 167 ms. Each trial was terminated either after 30 seconds or whenever the sheep was caught, whichever came first.

Each participant completed 14 trials of human sheep vs. computer wolf and 28 trials of human sheep vs. human wolf (half of which they played as the sheep and the other half wolf). This gave us a total of 140 trials in the human sheep vs. computer wolf condition and 280 trials in the human sheep vs. human wolf condition. For each trial of the stimulus generation phase, we recorded the location information for each disc at each time point. We then recreated the trials in video format using QuickTime for use in Experiment 1 (in "sheep view"—i.e., where the sheep was green, and the wolf was white). We selected a total of 72 trials for the perception task phase—40 trials from the human-controlled sheep vs. computer-controlled wolf condition and 32 trials from human-controlled sheep vs. human-controlled wolf condition. We note that the uneven number of trials across the two conditions arose for two reasons. First, we attempted to balance the number of trials that were selected where the wolf did eventually catch the sheep and where the wolf did not catch the sheep. We also had the goal of ensuring that all trials lasted at least 6 seconds (as trials shorter than 6 seconds provided participants an insufficient amount of time to perceive patterns of chasing behavior) and that there was a reasonable spread in the total trial time (i.e., from 6 seconds to 30 seconds). However, for the human-controlled sheep vs. human-controlled wolf condition, there were very few trials where the wolf did not eventually catch the sheep, meaning we had fewer trials to select from overall than the human-controlled sheep vs. computer-controlled wolf condition.

Only trials longer than 6 seconds were selected, after which all trials were cut down to between 5 and 15 seconds. Every trial was truncated by at least one second or more to avoid it being the case that the closest disc to the sheep at the end of each video was the wolf. For trials that lasted 30 seconds but were truncated to 15 seconds, we carefully examined each video and did not find evidence that either the wolf or the sheep changed their pattern of movements in the latter 15 seconds of the trial. Finally, when we created the videos for the trials of Experiment 1, we randomly labeled all the discs except the sheep disc with the numbers from 1–20 to allow participants to use the associated number when they were asked to indicate which disc they believed was the wolf.

## Perception phase

**Participants.** A total of 81 participants took part in the study in exchange for course credit after providing informed consent. Experimental procedures were approved by the University of Wisconsin—Madison Institutional Review Board.

The number of participants used in this study was determined using an a priori power analysis. We had the strong a priori belief that human-controlled wolves would be more difficult to detect and identify than computer-controlled wolves. As such, we started with an estimated effect size of at least moderate ($d = 0.5$). We then used G*Power [16] to determine the number of participants needed to achieve a power of 0.95. To achieve such power, fifty-four participants were needed. However, given the novelty of the design, we chose to exceed this value given the possibility that the effect would be in the small-to-moderate range instead. As we will see below, the actual effect size was, in fact, much larger than moderate ($d = 1.35$).

**Apparatus and display.** The trials from the stimulus generation study were then translated into video format and put onto the online platform Qualtrics for the current study. Participants completed the experiment using their own computers.

**Stimuli.** During each of the 72 trials of the study, the participants' task was to identify which of the white discs was chasing the sheep (i.e., to identify the wolf). Participants were only asked to respond at the end of the given movie. At the end of the trial, participants were asked to enter the number of the labeled disc that they believed was the wolf (or to guess if they weren't sure). No feedback was provided.

## Results

**\*Note\*** It has recently been argued that null results are often presented in the psychological literature in a manner that doesn't allow readers to make inferences regarding the strength of the nulls (given that frequentist statistics are not built to quantify these outcomes, [17]). Therefore, in order to further aid in the interpretation of null results in all experiments, we computed and report Bayes' Factors (*log(BF10)*) for all tests that were non-significant (note: Bayes' factors here are reported on a log 10 scale; for example, a *log(BF10)* of –0.6 indicates that there is about four times as much evidence for the null than the alternative).

### Are participants able to identify the computer-controlled wolf and the human-controlled wolf and if so, is performance in these conditions equivalent?

Our first question was simply whether participants could identify the wolf at better than chance levels in both conditions. Based on one-sample t-tests comparing participants' wolf identification rate to chance level performance (5%), this was indeed the case, $t(80) = 16.24$, $p < .001$, $d = 1.80$ for human sheep vs. computer wolf, $t(80) = 14.89$, $p < .001$, $d = 1.66$ for human sheep vs. human wolf.

We next compared performance across the two types of wolves. As expected, we found that participants were indeed better at identifying the computer-controlled wolf than human-controlled wolf (paired t-test; computer controlled wolf: $M = 46\%$, $SD = 23\%$; human-controlled wolf: $M = 27\%$, $SD = 13\%$, $t(80) = 12.18$, $p < .001$, $d = 1.33$, Fig 2).

## Discussion

Although participants were clearly less capable of identifying the wolf when controlled by a human being as compared to when the wolf was controlled by a direct heat seeking computer algorithm, they were nonetheless still able to do so well above chance levels. Because humans likely used a variety of strategies when playing the role of the wolf, in Experiment 2 we sought to examine, in a more controlled manner, whether multiple different algorithms that instantiated the chasing intention were equivalently easy to perceive.

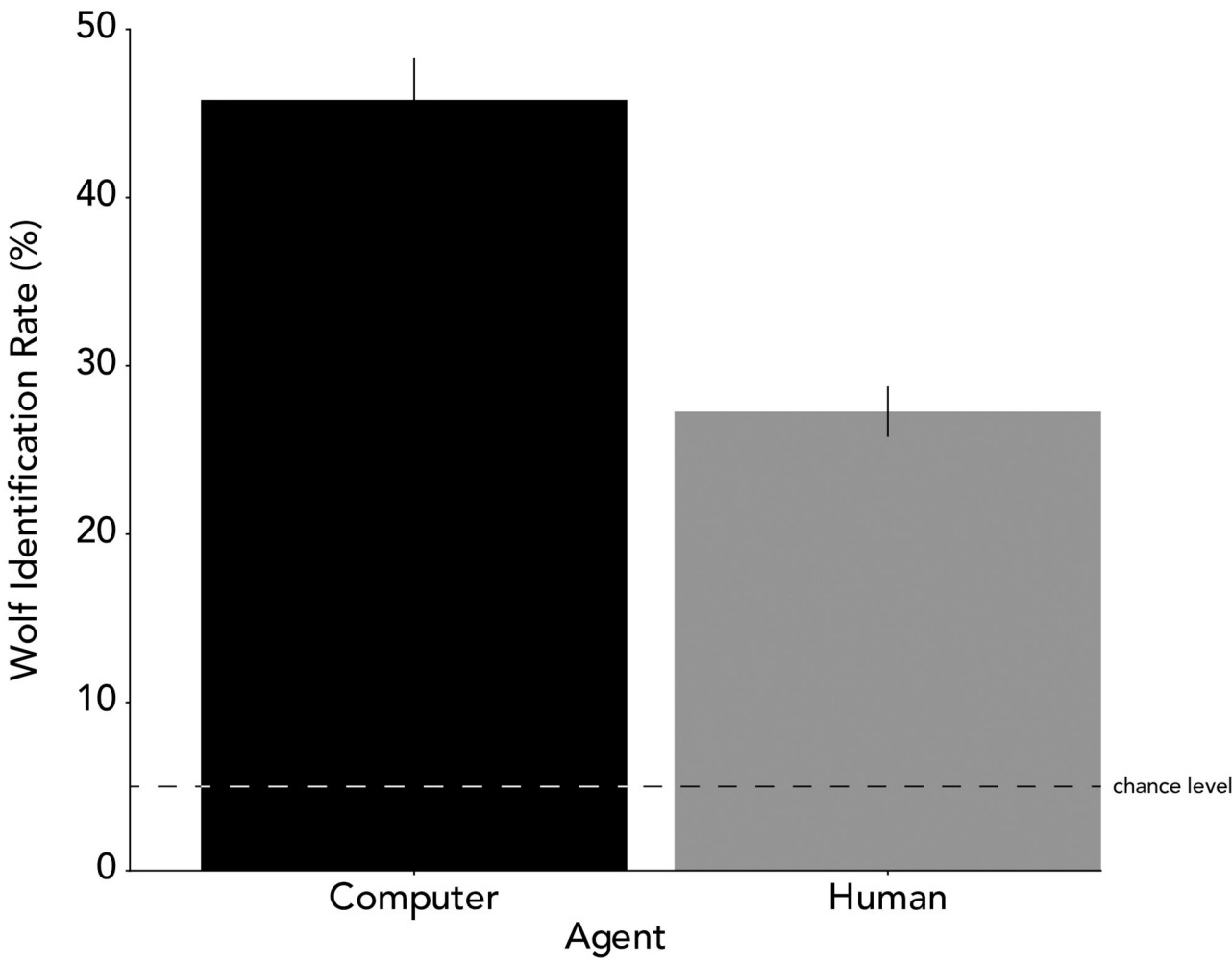

**Fig 2. Identification accuracy of computer wolf and human wolf trials.** Error bars represent the standard error of the mean. The dashed line represents chance level performance. Participants' wolf identification rate in both conditions was significantly better than chance. However, the wolf identification rate was significantly higher in the computer wolf than in the human wolf condition.

## Experiment 2: Detecting a chasing agent guided by different chasing algorithms

In Experiment 1, we saw that while participants were able to identify both a human-controlled and a computer-controlled (direct chasing) wolf at above chance levels, performance was markedly worse in the human controlled wolf condition. One possible reason for this difference is that human wolves used strategies other than direct chasing that were, in turn, more difficult to detect. In practice though, most human wolves did not use easily identifiable pure strategies, meaning that to address the question of whether certain pure chasing strategies are less detectable than others, a more controlled setup was necessary. Experiment 2 had three primary goals. The first was to compare participants' ability to identify a chasing agent controlled by multiple possible algorithms. Previous work in this space has largely used either hand-crafted stimuli (i.e., hand animated as in Heider & Simmel, 1944) or else simple chasing algorithms that utilized a single direct chasing strategy [11]. Here we sought to expand the space of algorithms that are consistent with the intention of chasing to include not only direct chasing

(i.e., at each time point move toward the sheep's location), but also sheep interception (i.e., at each time point move toward the point in the sheep's projected future path that would intercept the sheep) and sheep mimicking (i.e., perform the same behaviors as the sheep). The latter two algorithms were chosen for several reasons. First, those two types of chasing behaviors, along with the heat-seeking chasing that were previously tested in Gao and Scholl's experiment, were most often seen in Experiment 1's stimulus generation phase [12]. Second, they are plausible approaches that an intentional chasing agent could utilize to reach the goal of catching the sheep. In particular, while the sheep-mimicking condition might be of more limited utility in an unbounded environment (e.g., an open-plain), it has reasonable utility in the context of a bounded environment, which is obviously the type of environment that our research, like all previous work has considered. Notably this type of bounded environment does have potential analogs in the real-world—both due to man-made constructions (e.g., chasing within buildings, within fences, in dense cities, etc.), but also in the natural world (e.g., where environmental features like rivers, lakes, mountains, cliffs, etc. create boundaries).

Our second goal was to examine the impact of the number of distracting items in the display. It could be that intentional movements "pop out" from amongst background motion. If this is the case, increasing the number of distractors would have minimal impact on wolf identification rate. Conversely, if identifying intentional chasing involves some manner of serial search/comparison process, increasing the number of distractors should degrade wolf identification rate.

Finally, we sought to rule out one possible explanation for participants' ability to detect the wolf—which is that the wolf's movement is more strongly correlated with that of the sheep than any other disc in the display. Because simply moving in a correlated manner does not necessarily indicate chasing, in each display we included an actual chasing wolf (i.e., a wolf whose algorithm was meant to "catch" the sheep) as well as correlation-control wolf that chased a mirror-reversed version of the sheep (meaning that its movements were as correlated with the sheep's movement as the true wolf, but in such a way that it would not move toward/try to "catch" the sheep). If participants are using simple statistical relations between discs to judge wolf/sheep pairs, then they should be equally likely to select the true wolf and the correlation-control wolf. If on the other hand, they are using movements consistent with an intention to catch the sheep, they should essentially never select the correlation-control wolf.

## Methods

### Participants

A total of 42 new participants took part in the study in exchange for course credit after providing informed consent. Experimental procedures were approved by the University of Wisconsin—Madison Institutional Review Board. In this case, there was simply no existing data that would speak to the potential effect size. We thus chose to utilize an intermediate sample size between our estimated needed fifty-four in Experiment 1 and the number required given the extremely large observed effect in Experiment 1 (d = 1.35).

### Apparatus and display

Stimuli were presented on a 23-inch computer monitor. Participants were seated at a comfortable viewing distance of approximately 50 cm from the screen. All stimuli were generated using MATLAB (R2019a; MathWorks, Natick, MA) using the Psychophysics Toolbox 3 [13–15].

## Stimuli

All stimuli were presented within a black display frame subtending 32° by 24° of visual angle (calculated using a viewing distance of 50cm, although precise seating distance was not enforced in the experiment). Four different types of stimuli were present on each trial: the sheep, the wolf, normal distractors, and a correlation-control distractor wolf.

The sheep, wolf, and distractors were the same colors and had the same movement speed as in Experiment 1 above. Several changes though were made to the general setup as compared to Experiment 1. First, the sheep was no longer controlled by a human being. It was instead controlled by a fixed algorithm in which the sheep continuously moved around the perimeter of the display frame ("circled") throughout the trial with the direction of the motion being randomized across trials (i.e., on some trials it would circle in a clockwise manner and other trials a counterclockwise manner). A circling algorithm was chosen for the sheep based upon pilot data collected from human participants playing the role of sheep, which suggested that constant circling was one of the more common movement patterns.

The wolf's movement was controlled by one of three chasing algorithms on separate trials: direct chasing, sheep mimicking, and sheep interception (Fig 3). The direct chasing algorithm was the same as described in Experiment 1 above. In the sheep interception condition, the sheep's future trajectory (i.e., from its position and the direction of its last movement) was extrapolated and the wolf's target was then set in such a way to best "intercept" the sheep. Given that the display had "walls", there were not necessarily always linear interception points (i.e., the sheep would hit a wall before the wolf could intercept the sheep). As such, if the sheep was close to a wall/corner, the predicted trajectory included the expectation that the sheep would turn at that corner and move in the corresponding direction after turning. As with the sheep mimicking condition, in the interception condition, when the sheep was within 6° of the wolf, the wolf would convert to the direct chasing algorithm in order to catch the sheep. In the sheep mimicking condition, the wolf mimicked the sheep's behavior when the sheep was more than 6° away (e.g., if the sheep moved left, the wolf also moved left), and then switched to direct chasing if the sheep was closer than 6° apart.

Next, to minimize the difference in movement profiles between the normal distractors and the wolf, rather than utilizing a random motion for the distractors as in Experiment 1, the

## Wolf Types

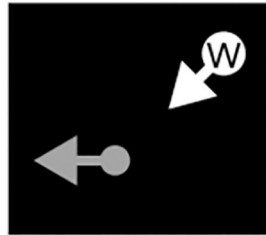 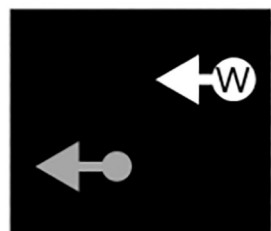 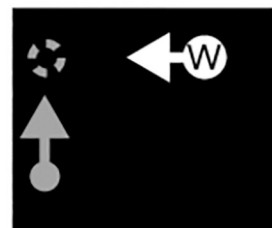

Direct Chasing Sheep Mimicking Sheep Interception

**Fig 3. Illustration of different chasing algorithms for Experiment 2–4.** In the direct chasing condition, the wolf always moved toward where the sheep was located. In the sheep mimicking condition, the wolf mimicked the sheep's behavior unless they were close to the sheep, then the wolf switched to the direct chasing algorithm. In the sheep interception condition, the wolf predicted the sheep's movement trajectory and moved toward the predicted location unless they were close to the sheep, then the wolf switched to the direct chasing algorithm.

distractor motion was generated by having each distractor chase a randomly generated "invisible sheep" throughout the trial (using the direct chasing algorithm). The "invisible sheep" moved in a similar fashion compared to the actual sheep (i.e., circling around the background), but unlike the actual sheep, they could be initialized anywhere in the background rather than being restricted to just the outer perimeter of the background. If the "invisible sheep" was caught by its corresponding normal distractor, a new "invisible sheep" was randomly generated within the background and the corresponding distractor would continue to chase the newly generated "invisible sheep". Half of the trials (equally distributed across chasing algorithms) had 19 distractors and the other half had 10.

Finally, we included a "correlation-control" distractor that was the same color, size, and had the same speed as the other distractors (and the wolf). This disc chased a mirror image of the actual sheep (mirrored around the center of the background). On all trials, this distractor used the same chasing algorithm used by the actual wolf on the given trial (i.e., in the sheep mimicking condition, this distractor would also use the same mimicking algorithm to chase the mirrored sheep). This distractor was included so that there was always one disc in the display that had almost perfect motion correlation with the actual sheep even though it did not engage in the chasing process.

32 trials were generated for each condition (i.e., direct chasing, sheep interception, and sheep mimicking) for a total of 96 trials in all. The one constraint during generation was that only trials where the wolf had not caught the sheep within the first 11 seconds were selected. We then subsequently cut all the to-be-displayed movies of the trial down to 10 seconds. This reduced the extent to which the participants could simply always indicate that the wolf was the disc closest to the sheep at the end of the recording.

## Procedure

Experimenters read the task instructions to participants. Participants were then trained using 6 practice trials (2 of each chasing type) to ensure they understood the task—which was, as in Experiment 1, to identify which of the white dots was the wolf. No feedback was provided. The results of these practice trials were discarded and not analyzed. The 96 total trials were divided into blocks of 24 trials such that there were an equal number of trials from each condition to ensure that the trial types were equivalently distributed throughout time. Participants first completed 48 trials of the task, then completed a secondary task not analyzed here in which participants actively controlled the sheep to avoid being caught by the wolf, before completing the second set of 48 trials of the task. Given that the emphasis of the current work is on the perception of intentionality, the active sheep-control task is not a major consideration and will not be further considered here. Yet, although this task is not of relevance to the current work, it is nonetheless the case that this intervening task could have influenced participants' perceptions. As such, trials completed after the active sheep-control task were not included in the current analyses. The whole experiment took around 1 hour, and participants were asked to take mandatory breaks after every two blocks to reduce fatigue and eye strain.

## Results

### Can participants identify visual signals of intentionality under different chasing algorithms?

Our first goal was to compare participants' ability to identify a chasing agent controlled by multiple possible algorithms. All the algorithms were designed in such a way that they would, in fact, catch the computer-controlled sheep. Based on a simulation of the three algorithms,

the sheep was successfully caught 37.6% of the time in the direct chasing condition, 99.35% in the sheep mimicking condition, and 84.93% in the sheep interception condition. The discrepancy in success rate is largely due to the difference in speed between the sheep and the wolf (i.e., the sheep is faster than the wolf and thus a direct-chasing algorithm will frequently "miss" the sheep as it moves past). However, given that previous work has utilized such a direct-chasing algorithm, we felt this imbalance of catching success was acceptable as it would, if anything, be expected to favor the new algorithms (since those, by their greater success rate, were unequivocally trying to catch the sheep).

While wolf identification rate was well-above chance for all algorithms for both the 10- and 19-distractor versions, $t(41) = 30.51$, $p < .001$, $d = 4.71$ for the 10 distractors condition, $t(41) = 19.71$, $p < .001$, $d = 3.04$ for the 19 distractors condition (note that chance level in this task is 5% for the 19 distractors condition and 9% for the 10 distractors condition), the wolf identification rate was not equivalent across conditions. In a 3 (condition) x 2 (distractor) ANOVA, the main effect of chasing algorithm condition was significant, $F(2,246) = 46.43$, $p < .001$, $\eta_p^2 = 0.27$, with participants' wolf identification rate being best in the direct chasing condition, followed by the sheep interception and lastly the sheep mimicking condition (confirmed by post-hoc tests; Fig 4).

### Does the number of distractors affect participants' ability to identify visual signals of intentionality?

Our second goal was to assess whether wolf identification rate changed as a function of the number of distractor discs. In the same 3 (condition) x 2 (distractor) ANOVA analysis discussed above, we found a significant effect of the number of distractors presented on the screen with better identification rate when there were fewer distractors (Fig 4), $F(1,246) = 108.10$, $p < .001$, $\eta_p^2 = 0.31$. We did not find a significant interaction between condition and distractor, $F(2,246) = 0.61$, $p = .54$, $\eta_p^2 < .001$. Given the non-significant result, we further report the BF, which was $log(BF10) = -0.90$.

### Do correlated movements indicate intentional behaviors?

Our third and last goal was to rule out the possibility that the perception of chaser/chasee pairs were determined entirely by statistical regularities (i.e., correlations between the movements). In all trials, there was one distractor that was chasing an invisible mirrored version of the actual sheep and thus had the same correlation in movement pattern with the sheep as did the true wolf. If participants were using the co-variation in movement as a strong cue, they should, at a minimum, disproportionately select the correlation-control distractor more often than other distractors. We analyzed whether participants chose this distractor over other distractors in trials they got wrong. Based on a binomial test, we found that participants chose this distractor (i.e., the wolf chasing the mirrored sheep) much less often (0.55%; note that chance level in this case would be around 5% for the 19-distractor condition and 9% for the 10-distractor condition) compared to other distractors when participants chose discs other than the real wolf, $p < .001$.

### Discussion

We implemented different types of chasing algorithms to examine whether participants could equivalently identify chasing agents driven by different types of chasing algorithms. While participants performed above chance in all conditions, they were not equivalently able to identify the wolf in each. Instead, participants performed best in the direct-chasing condition and

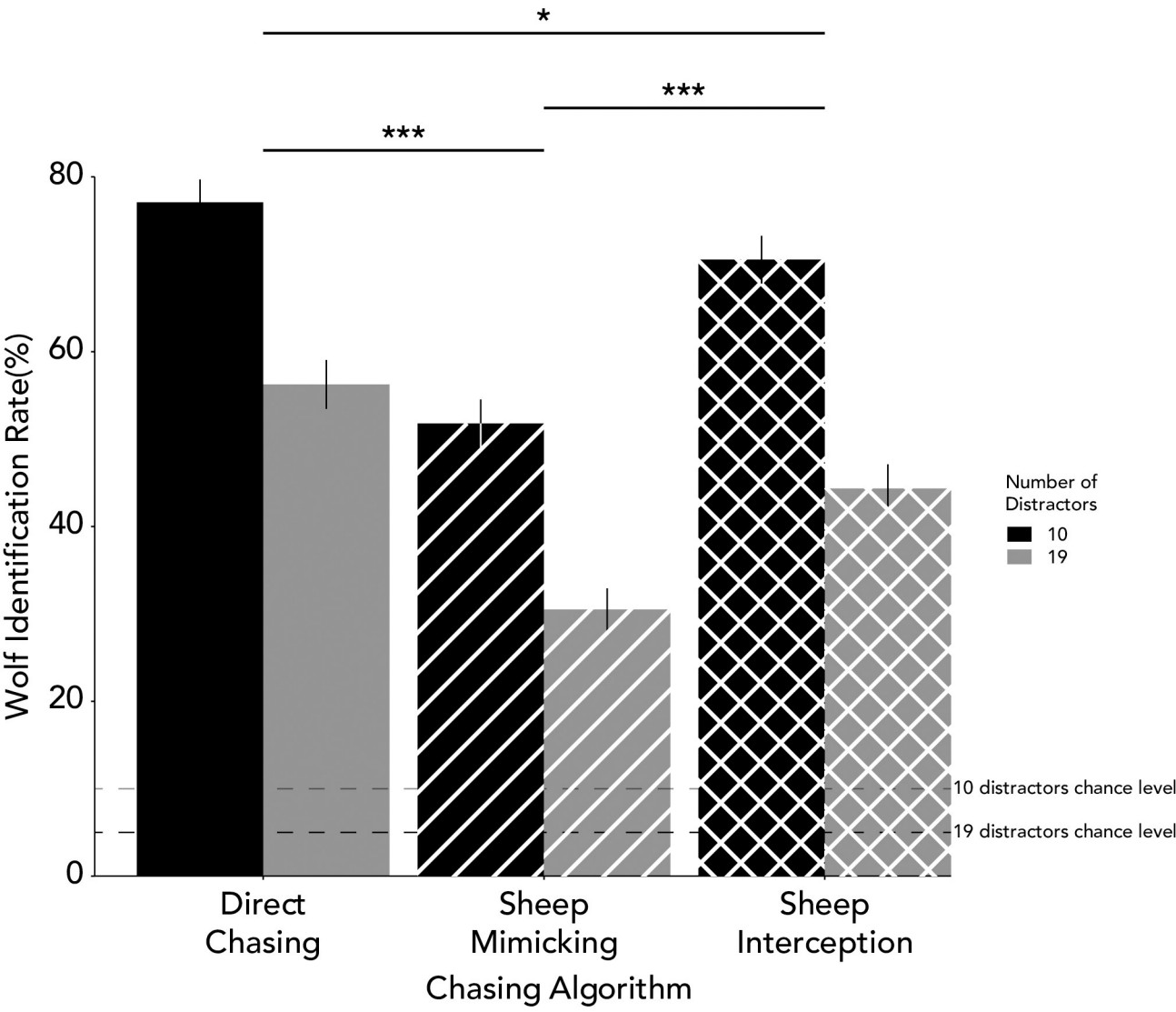

**Fig 4. Identification accuracy across different chasing algorithms.** Each bar represents the identification accuracy of a chasing algorithm and two distractor conditions. Error bars represent the standard error of mean. Dashed lines represent chance level performance in 10 and 19 distractors conditions. Asterisks above bars indicate significance level (*: < .05; **: < .01; ***: < .001). Participants' wolf identification rate was best in the direct chasing condition, followed by the sheep interception, and lastly the sheep mimicking. Participants were also better in the 10 distractors condition compared to the 19 distractors condition.

worst in the sheep-mimicking condition. We note that this is the exact opposite pattern with respect to the three types of wolves' ability to actually catch the sheep (where the direct-chasing wolf was least successful, the sheep-intercept wolf was next, and finally the sheep-mimicking wolf was nearly 100% successful at catching the sheep). The reduced wolf identification rate in the sheep mimicking condition could suggest that either proximity or perhaps consistently increasing proximity (i.e., reducing the distance to the target) could be an important cue to the chasing intention. The mimicking wolf kept a more or less equal distance from the sheep until one or the other was constrained by a wall. This would be consistent with results from Gao & Scholl [12] who manipulated the percentage of time the wolf spent chasing the sheep. When the wolf spent less time directly chasing (and more time moving randomly), it caused a

significant reduction in participants' ability to escape the wolf. This was described as "stalking." The wolf in both the direct chasing and the sheep interception conditions were constantly increasing proximity which, consistent with that view, made it easier for participants to identify the wolf. We also tested whether perception of chasing evoked a "pop-out" effect by introducing different densities of stimuli. The greatly reduced wolf identification rate in the 19 distractors condition suggests that participants might be engaging in sequential search. Lastly, we aimed to test whether correlated movements indicate intentional behaviors by including a wolf that is chasing the mirror image of the sheep in which it showed almost perfect correlated motion but did not engage in actual chasing. We did not find that participants chose the wolf chasing the mirrored sheep more often than other distractors. In fact, they chose that particular disc far less often, again suggesting that proximity may be an important cue since the mirror-chasing wolf almost always maintained a large distance from the sheep even though it had almost perfect correlation with the sheep's movement.

## Experiment 3: Detection of both the chasing agent and the agent being chased

In Experiment 2 we found that participants were capable (although not equivalently capable) of identifying wolves whose movements were generated by three different chasing algorithms. Yet, because previous work on the perception of chasing has nearly always focused on participants' abilities with respect to the chaser (i.e., to detect, identify, or avoid the wolf; [11, 12]), it is unclear whether both members of the chaser/chasee pair are equivalently identifiable. Given the work suggesting that patterns of relational motion between the wolf and sheep are critical in identifying chasing, one may surmise that detection of the two types of agents should be equivalent (i.e., if one has detected that there are relational movement patterns between two discs consistent with chasing, one must know the identity of the two discs). On the other hand, it is potentially more ecologically important to detect a chasing agent rather than an agent being chased (e.g., if there was a wolf chasing a herd of sheep, it's less important to know which particular sheep is being chased than that the wolf is chasing a sheep). Furthermore, while it is not necessarily the case that a chasing agent *must* have intentionality (e.g., a weaker magnet being pulled toward a stronger magnet), it is arguably the case that there are more cases where the chaser has intentionality while the chasee does not, rather than the opposite case (e.g., if a ball on a string is tied to the collar of a dog). Thus, it could be that the ability to detect the wolf could exceed that of detecting the sheep (although we do note that one could also make the counter-argument that humans have frequently acted as chasers, often in concert with other humans, and when doing so may focus on a single prey animal; in that case, detecting which prey animal is being chased by the "pack" may be more important than knowing who the chaser is). To adjudicate between these possibilities, we thus had participants perform a similar identification task as in previous work (and Experiment 2). However, rather than only being asked to identify the wolf, in separate blocks of trials they were asked to identify either the wolf or the sheep.

## Methods

### Participants

A set of 40 new participants took part in the study in exchange for course credit after providing informed consent. Experimental procedures were approved by the University of Wisconsin—Madison Institutional Review Board. Here our sample size was chosen based upon the significant results seen in Experiment 2 with N~40.

### Apparatus and display

The apparatus was exactly the same as in Experiment 2.

### Stimuli

**Wolf identification task.** All aspects of the stimuli generation were the same as in Experiment 2 (72 out of the 96 trials used in Experiment 2 were selected; 24 per chasing condition).

**Sheep identification task.** All aspects of the stimuli generation were the same as in Experiment 2 except the following: In this task, instead of asking the participants to identify which disc was the wolf given the sheep's movement trajectory, we asked participants to identify which disc was the sheep given the wolf's movement pattern. Because the sheep's movement speed in the original passive observation task was faster than the distractors and the wolf, this speed difference would immediately serve to identify the sheep. We thus had to generate new sets of trials to compensate for the speed difference. While the distractors' speed remained the same (9.6°/s) as in the wolf identification task, we modified the sheep's speed to be the same as the distractors (9.6°/s) as well as reducing the wolf's speed to ⅔ of the sheep's speed (6.4°/s; which is the same ratio in the wolf identification task). The sheep was drawn as a white disc that had the exact same appearance as all other distractors. The wolf was drawn as a green disc. We instructed the participants to focus on the wolf and told them that the wolf is chasing a sheep that is hiding among other discs.

Trials for the sheep identification task (72 trials) were generated via a similar procedure as described for Experiment 2. The same randomization process (i.e., blocked to ensure equal distribution of chasing algorithms/distractor numbers) was used as in Experiment 2.

**Procedure.** Participants completed the two tasks in an interleaved fashion: they first completed one of the two tasks (i.e., sheep identification or wolf identification) in a first block of 36 trials, followed by the other task in the second block (36 trials), then switched back to the first task in the third block (36 trials), and finally the other task in the fourth block (36 trials; so in all wolf-sheep-wolf-sheep or sheep-wolf-sheep-wolf). The order of the tasks as well as which set of trials was run first were counterbalanced across participants. The experiment took around 45 minutes and participants were instructed to take a mandatory break after the second block to reduce fatigue and eye strain.

## Results

### Are identification rates equivalent across the wolf identification and sheep identification tasks?

Our goal was to investigate whether participants showed equivalent identification rates when they were tasked to detect either the agent chasing, or the agent being chased. Based on one sample t-tests, the identification rate of both the wolf identification task ($M$ = 57%, $SD$ = 10%) and the sheep identification task ($M$ = 72%, $SD$ = 19%) were above chance for both distractor conditions, all $p$s < .001, all $d$s > 2.30.

We next ran a linear mixed effects model with task, chasing algorithm, and number of distractors as fixed effects and found that there were significant main effects of task (sheep identification vs. wolf identification), $F(1,39)$ = 114.58, $p$ < .001, $\eta_p^2$ = 0.21, chasing algorithm, $F(2,39)$ = 10.92, $p$ < .001, $\eta_p^2$ = 0.02, and number of distractors, $F(1,39)$ = 193.95, $p$ < .001, $\eta_p^2$ = 0.31. More specifically, participants' sheep identification rate was much better across the board compared to their wolf identification rate (Fig 5). We did not find any significant interactions between any of the fixed effects (all $p$s > .05).

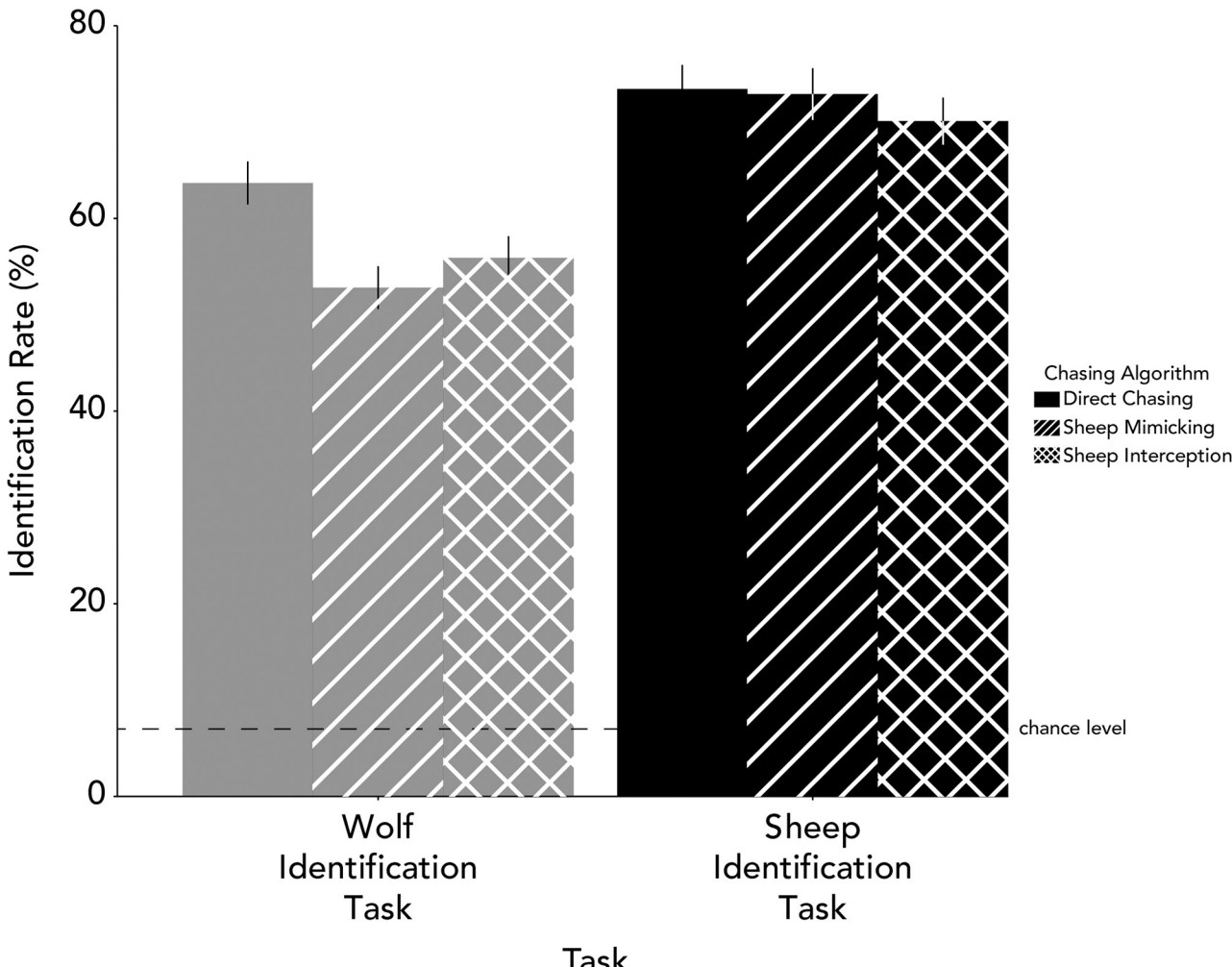

**Fig 5. Identification accuracy in the wolf identification task and sheep identification task.** Each bar represents one chasing algorithm (from left to right: direct chasing, sheep mimicking, sheep interception). Error bars represent standard error of mean. Dashed line represents the average chance level performance across 10 and 19 distractors conditions. Participants showed a better identification rate in the sheep identification task compared to the wolf identification task across the board.

## Discussion

We investigated whether participants showed equivalent identification rates when they were asked to identify the agent chasing as well as the agent being chased. Contrary to our expectations participants performed better in the sheep detection task than in the wolf detection task. There are several possible reasons for this result, including differential cue use or that certain cues are stronger in the sheep as compared to the wolf. In particular, the sheep was using a strongly deterministic algorithm (circling) that was potentially very salient amongst the distractors. These possibilities will be returned to in the conclusions.

## Experiment 4: Detection of chasing agent when the sheep is invisible

As noted above, while previous studies have suggested that correlated movements between sheep and wolf are necessary to induce the perception of chasing [11, 12] because

manipulations of this correlation have always involved altering the movement patterns of the wolf (i.e., making some of the wolf's movements random or less directed toward the sheep), it leaves open the possibility that it is the contrast of the wolf's movement against the random movement of the background that is necessary to evoke the perception of chasing, rather than requiring both sheep and wolf. Though the chasing absent condition has been explored in previous studies [11], the participants' goal in this previous work was to determine whether chasing was present or not. It thus did not answer the question of whether participants could detect the difference between intentional behaviors vs. non-intentional behaviors absent the sheep. As such, to establish that both the chaser and the chasee are required for participants to detect chasing, we utilized a very similar setup as in Experiment 2, but in this case made the sheep invisible. Our expectation was that participants in this condition would not be able to identify the wolf above chance.

## Methods

### Participants

A set of 45 new participants took part in the study in exchange for course credit after providing informed consent. Experimental procedures were approved by the University of Wisconsin—Madison Institutional Review Board.

### Apparatus and display

All apparatus and display were exactly the same as in Experiment 2.

### Stimuli

All stimuli were the same as in Experiment 2 except the sheep was made invisible.

### Procedure

All procedures were the same as in Experiment 2.

## Results

### Can participants still identify the wolf without the sheep?

Our goal was to investigate whether both the chaser and the chasee are required for participants to detect chasing. Based on a two-tailed one sample t-test comparing participants' wolf identification rate to chance level (5% for 19 distractors condition and 9% for 10 distractors condition), participants' wolf identification rate was not significantly greater than chance levels in either the 10 distractors condition, $t(44) = -2.49$, $p = .01$, $d = -0.37$, or the 19 distractors condition, $t(44) = -1.02$, $p = .31$, $d = -0.15$ (Fig 6). Note that for the 10 distractors condition, participants actually performed significantly below chance level. For the 19 distractors condition, we found a $log(BF10)$ of -0.58, indicating almost four times as much evidence for the null than the alternative (i.e., participants were four times more likely to perform at chance level than different from it). For the 10 distractors condition, we found a $log(BF10)$ of 0.41, indicating around 3 time as much evidence for the alternative than the null (i.e., again noting the direction—participants were three times more likely to perform worse than chance levels).

## Discussion

We made the sheep invisible in this task to investigate whether participants required both the chaser and the chasee to identify a chasing agent. We found participants' wolf identification

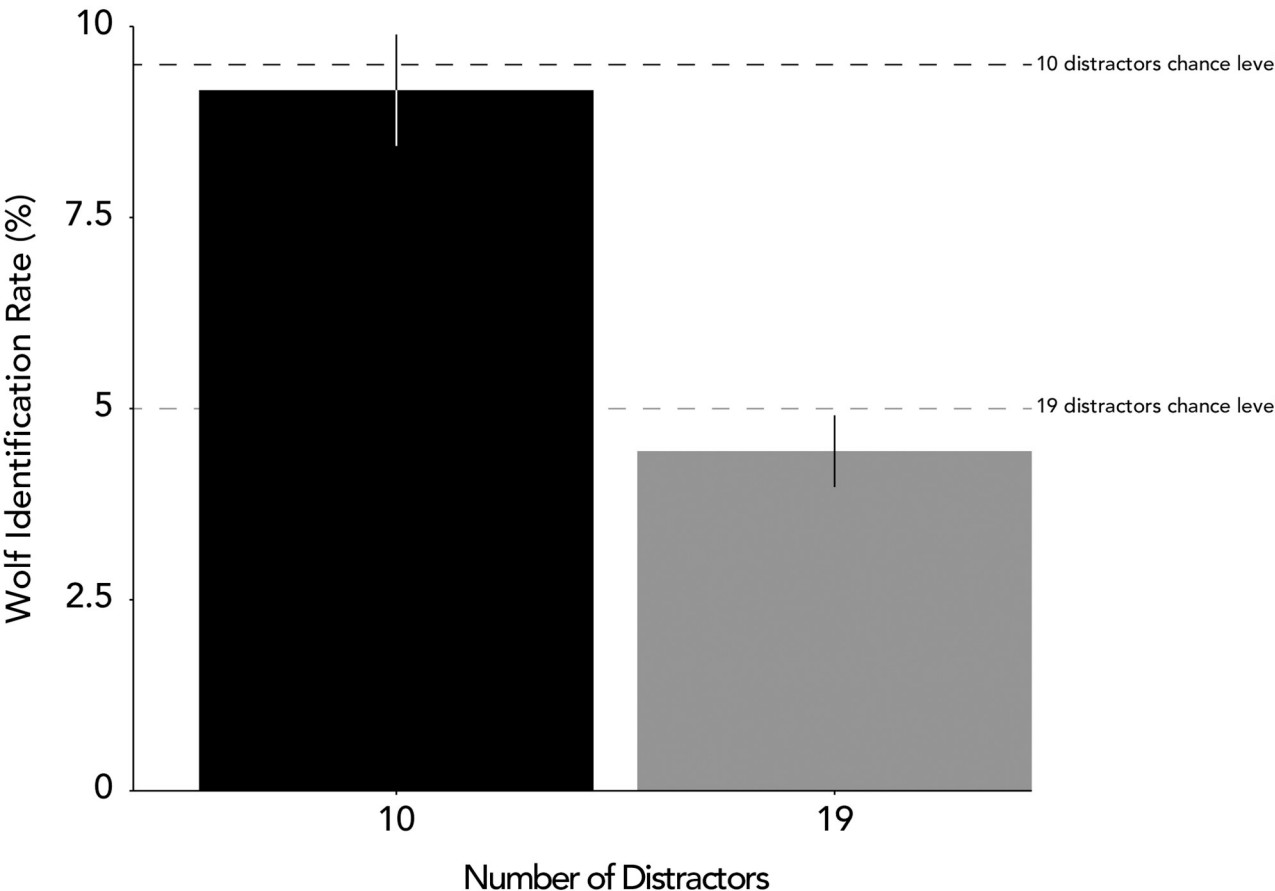

**Fig 6. Identification accuracy separated by the number of distractor discs.** Horizontal dashed lines represent chance level identification rate. Participants' wolf identification rate was at chance in both the 10 and 19 distractors conditions.

rate was at or below chance levels for both the 19 and 10 distractor conditions. This suggests that people do not just rely on the wolf's movement pattern to identify visual signals of intentionality but rather the relationship between the wolf and the sheep.

## General discussion

The results from the four experiments presented here are consistent with previous research showing that our visual system is sensitive to intentionality, especially in a chasing paradigm [11, 12]. We have shown that the perception of intentionality occurs for different types of chasing agents (human and computer) and for different chasing algorithms (such as direct chasing, mimicry, and interception). However, we have also demonstrated that perception of intentionality is differentially affected by these factors, suggesting that perceiving the intentionality of real-world agents may be hindered by the use of more complex strategies (which might allow agents to mask their intentionality to avoid detection). Overall, and consistent with previous research, we found that participants were least able to detect chasing when the wolf and sheep exhibited more variable movement patterns, such as was the case when the wolf was controlled by another human rather than a deterministic algorithm. In contrast, participants were most able to detect chasing when the chasing agent's movement was constantly directed at the agent being chased (i.e., heat-seeking) as compared to chasing patterns that predicted or mimicked the chasee's behavior.

The sheep mimicking condition was of particular note as it was the most difficult for participants to identify. In this algorithm, the wolf would often move away from the sheep (e.g., if the sheep was on the right side of the wolf moving left, the wolf would also move left). These results are thus similar to those found by Meyerhoff and colleagues [18] who also found that changes in inter-object spacing are an important cue to intentionality. Similarly, Takahashi and Watanabe [19] found that synchronous motion weakened the subjective ratings of perceived animacy, which would also be true of a wolf that was mimicking the sheep's behavior.

Moreover, the sheep mimicking algorithm was designed in a way that would keep the wolf moving around an edge or corner if the wolf was already at the edge/corner of the background and had to keep moving in that direction based on the sheep's movement (e.g., the wolf is at the left edge of the background and the sheep is moving left). This moving around the edge/corner behavior made it even less obvious that the wolf was chasing the sheep. Interestingly, at least anecdotally, we observed humans exhibiting at least qualitatively similar behaviors (e.g., in the recordings we used for Experiment 1) which essentially resembles a "waiting for the sheep to approach" strategy. Consistent with Gao and Scholl's [12] interpretation, participants were likely treating these cases as negative evidence (i.e., evidence against chasing) that in turn prevented them from successfully identifying the chasing agent. While we are unaware of any work that has explicitly used a "race-model" formulation for this task, our results are broadly consistent with such a view. For instance, if the mimicking condition provides "negative evidence" of chasing, then it would be more likely for a distractor to "win" the race to be considered the wolf. The addition of more distractors would likewise increase the probability of a distractor being selected as the wolf. Finally, eliminating one of the chaser/chase dyad would reduce the rate at which evidence accumulates and thus increase the chance of a distractor being erroneously labeled the chaser.

## Limitations and future directions

Though we used three different types of computer algorithms to control the movement of the chaser, there obviously exists a potentially infinite space of such algorithms (though not necessarily with equivalent catch-success rates). For example, in our pilot work, we found human wolves often moved to a location where they could "wait" (ambush style) for the sheep to approach. That behavior is both highly intentional, but would not be easily detected by most "chasing"-motion templates. As such, considering an even broader range of algorithms that are intentional would be valuable for future work. In this vein, chasing is also only one of many examples of intentional behavior. Other goal-directed behaviors, such as repulsion, avoidance, or attraction may involve quite different motion pattern templates that would be valuable to investigate.

Future work could also extend our current results in a number of ways. For instance, in Experiment 3, the algorithm used by the sheep was simplistic (i.e., circling). While the circling algorithm was not a strawman, but was instead chosen based on pilot data indicating that this was a common strategy employed by human participants playing the role of sheep, it's rudimentary nature may have nonetheless played a major role in determining the perceptual outcomes (as compared to, for instance, a sheep that tried to move randomly to avoid a wolf). Future studies should include more escape strategies used by the sheep. For example, in the human-controlled sheep/wolf condition, participants were observed using strategies other than circling (though, again, many were somewhat simplistic—such as moving the sheep along one edge of the background and only switching to other edges occasionally). Including different algorithms used by the sheep might give us the ability to answer why people were better at detecting the agent being chased. While we hesitate to read too deeply into the patterns,

numerically the most difficult to detect sheep was in the context of a wolf that was utilizing a trajectory intercept strategy. This would make sense from base principles if the perception of the chaser/chase relationship is inherently about a dyad (i.e., if things like proximity and correlated motion cues matter for detecting the chaser, since if those are weaker, it *should* also impact the ability to perceive the chasee). More titrated sheep/wolf pairs would be needed though to address this issue more conclusively.

## Acknowledgments

We would like to thank SF, OF, LV, TR, AK, SL, DO, NB, WJ, AS, for assisting in recruitment and data collection.

## Author Contributions

**Conceptualization:** Mohan Ji, Emily J. Ward, C. Shawn Green.

**Data curation:** Mohan Ji.

**Formal analysis:** Mohan Ji.

**Investigation:** Mohan Ji.

**Methodology:** Mohan Ji.

**Project administration:** Mohan Ji.

**Software:** Mohan Ji.

**Supervision:** Mohan Ji, Emily J. Ward, C. Shawn Green.

**Validation:** Mohan Ji.

**Visualization:** Mohan Ji.

**Writing – original draft:** Mohan Ji, Emily J. Ward, C. Shawn Green.

**Writing – review & editing:** Mohan Ji, Emily J. Ward, C. Shawn Green.

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
