## [Decision Letter · Decision Letter 0]

10 Jan 2023

PONE-D-22-29587Realistic and complex visual chasing behaviors trigger the perception of intentionalityPLOS ONE

Dear Dr. Ji,

Thank you for submitting your manuscript to PLOS ONE. After careful consideration, we feel that it has merit but does not fully meet PLOS ONE’s publication criteria as it currently stands. Therefore, we invite you to submit a revised version of the manuscript that addresses the points raised during the review process.

Two expert reviewers have assessed your work. Both reviewers agree that your manuscript is interesting and worthwhile. The reviewers also raise several points which you should address, primarily regarding the choice of experimental conditions and analyses. Reviewer 2 suggests it may be appropriate to run an additional experiment, but this may not be necessary if you can better explain the purpose of the "mimicking" condition. In general, there are several points in which you should be able to respond to the reviewer comments by clarifying your reasoning in the manuscript. Overall, I think you should be able to address all reviewer comments, and I look forward to receiving your revised work. 

We look forward to receiving your revised manuscript.

Kind regards,

Guido Maiello

Academic Editor

PLOS ONE

Journal Requirements:

Reviewers' comments:

Reviewer's Responses to Questions

**Comments to the Author**

1. Is the manuscript technically sound, and do the data support the conclusions?

Reviewer #1: Yes

Reviewer #2: Yes

2. Has the statistical analysis been performed appropriately and rigorously? 

Reviewer #1: Yes

Reviewer #2: Yes

3. Have the authors made all data underlying the findings in their manuscript fully available?

Reviewer #1: Yes

Reviewer #2: Yes

4. Is the manuscript presented in an intelligible fashion and written in standard English?

Reviewer #1: Yes

Reviewer #2: Yes

5. Review Comments to the Author

Reviewer #1: Well written and engaging manuscript. Authors present a systematic study of how humans perceive chasing from low level motion characteristics. The rationale for each step is appropriate and the methods chosen align with the research questions.

I have a few queries, although I see these queries as something that the authors may wish to consider to enhance the paper – not as fundamental flaws.

1. Why are both frequentist and Bayesian statistics reported? Frequentist and Bayesian statistics have fundamentally different philosophies that underlie them, and it is a little strange to double up. It is probably best to choose the statistical philosophy that aligns better with your own philosophy or is most appropriate for the questions that you are asking.

2. If you stick with frequentist statistics, I advise reporting effect sizes for all tests (there are many without). Sometimes it appears as though the BF is taking the place of the effect size because the BF factor appears where the effect size should go. To avoid confusion, it probably should be separated clearly so readers do not assume that the authors are trying to use the BF factor as the effect size of a frequentist calculation.

3. I had difficulty understanding the correlation-control wolf. It might be worth highlighting that the videos are available on the OSF and making example videos for the ones that are in matlab code (not everyone will have access to matlab given that it is not free).

4. I am note sure if ecologically It would be more important to identify a chasing agent (pg. 23). In many cases the opposite might be true. Consider hunting in a pack, which our evolutionary ancestors would have done (and one of our closest evolutionary ancestors do – chimpanzees), perhaps it is more important to identify the relationship between the chaser and chased? If one of your hunting group is focused on a particular individual prey you would be best to focus your efforts on that prey – then it becomes important to identify the relationship between the chaser and chased – without knowledge of both it will make it difficult to succeed (and communicative cues would alert the chased more). In other words, it is very important to know which sheep is being chased, otherwise group chasing strategy would be poor.

I sign all my reviews regardless of their content; be it a positive or negative evaluation. I aim to provide high quality, thorough, considerate and respectful reviews. Transparency assists in meeting those aims. Should this review not meet those aims I am always happy to take feedback on my review.

Sincerely,

Merryn Constable

Reviewer #2: Summary

Past research on the visual detection of chasing movements has found that direct (heatseeking) chasing is more detectable. The current research demonstrates that when two participants engage in "real" chasing, their movement is not nearly as heatseeking as the algorithm that has typically been studied. It also confirms in various ways the past finding that heatseeking chasing is more detectable than other types, while also finding some sensitivity to other kinds of chasing.

In Experiment 1, two subjects played a game against one another, in which they used the arrow keys to control dots, and one subject chased the other. The main contribution of the paper was to discover that, in contrast to the simple 'heatseeking' chasing algorithm used in previous research, subjects did not chase directly towards each other, but rather followed a pattern of indirect ('subtle') chasing. In part 2 of this experiment, a new group of subjects was asked to perform an 'identify the chaser' task involving either algorithmically heatseeking trajectories or recordings of the realistic chasing from part 1. Consistent with past research showing that chasing is less detectable when it is more indirect, subjects were more sensitive to heatseeking chasing than the more indirect realistic chasing.

In Experiment 2, accuracy identifying the wolf was again measured, comparing three algorithms — heatseeking, predictive chasing, and displays in which the wolf copied the sheep. The important discovery in this experiment was that performance was again best for heatseeking chasing, even though predictive chasing was arguably more efficient and more naturalistic.

In Experiment 3, the same chasing algorithms were varied, and subjects identified either the wolf or the sheep, and were better at detecting the sheep, probably because it moved in a more distinctive way by circling the perimeter of the display. (Also, one of the distractors chased the sheep's invisible location, and subjects were not likely to falsely identify this distractor as chasing. This is consistent with past claims that merely correlated movement of a dot chasing the invisible mirror image of another dot in the display does not look like chasing.)

In Experiment 4, subjects again failed to detect a wolf when it chased an invisible sheep.

Overall, this article was an interesting read, but could benefit from some revisions.

Big Points

In Experiment 2, comparing sensitivity to heatseeking vs. more naturalistic and effective "interception" chasing seems well motivated. But it is less clear what is learned from testing sensitivity to "mimicking" chasing. It seems unlikely that subjects in the task actually spontaneously saw this as chasing (maybe this is the authors' discovery, but why is that important?). So we would recommend rerunning Experiment 2 without that condition — which would also allow an opportunity to rerun that experiment without including an awkwardly unanalyzed task. Or perhaps the authors could better explain the purpose of including this condition.

We would also recommend removing Experiment 3, in which there was better detection of the sheep than the wolf, since the reason for this is almost certainly just a confound in the sheep's movement looking more distinctive.

Small Points

Looking at the videos, it seems that the distractors on average moved within a small area, whereas the wolf and sheep covered much longer distances, and were more peripheral. Across studies, could this have played a role in the patterns of results you observed?

Lines 56-65: Up to the authors, but it's possible that this section can be made more concise.

Line 129: Maybe change "two additional issues" to "two additional issues, which have been explored in past research", since past work on chasing has already studied both the effects of # of distractors and whether correlated movement of this sort is sufficient to see chasing.

Line 262: Not ideal that the experimenters had a hand in selecting/trimming the naturalistic chasing stimuli that got used for part 2. Without this curation, is it possible that the naturalistic chasing stimuli would have been detected as well as the heatseeking chasing stimuli in part 2?

Line 279: What was the stopping rule for data collection?

Line 334: It would be good to include more explanation of why these three particular algorithms were tested. The "copying" condition in particular seems unmotivated.

Line 434: Actively controlling the sheep to escape the wolf has been used as a measure of the perception of intentionality in the past, so not all readers will be convinced that this task as not relevant.

Line 464: Subjects in the wolf-mimics-sheep condition may have learned to respond to this stimulus, but just because they succeeded (albeit less so than when there was heatseeking or predictive chasing), does this mean that they really saw it as an intentional behavior? (Or is the claim that their failure to detect this was due to not seeing it as intentional?) How it looked to subjects seems like an important point, if one of the main claims of the article (e.g. on line 124) is that subjects' ability to detect different kinds of chasing above chance implies that they are perceiving the underlying cause/goal of the chasing rather than something more superficial about chasing movements.

Line 612: We recommend removing Experiment 3 from the paper altogether, since, as the authors acknowledge, the distinctive 'circling' pattern is almost certainly the whole explanation for why the sheep was more detectable.

Line 615-621: Haven't wolves chasing an invisible sheep commonly been used in past chasing detection experiments as a target-absent display? It might be better to say this outright, and to explain how Experiment 4 teaches us something new.

Line 719: Arguably this could explain the constant performance across the three conditions (e.g. if the sheep looked very distinctive in all three conditions on the basis of its very different movement, leading it to be quite detectable regardless of the wolf's style of chasing).

6. PLOS authors have the option to publish the peer review history of their article (what does this mean?). If published, this will include your full peer review and any attached files.

Reviewer #1: No

Reviewer #2: No

---

## [Author Response · Author response to Decision Letter 0]

9 Mar 2023

Response to Reviewers

We would like to thank the reviewer’s time and effort to comment on the manuscript. We have edited the manuscript to address their concerns and we believe the manuscript is much improved as a result. Find below responses to each concern raised by the reviewers (reviewer comment in black, our response in red. 

Reviewer #1: Well written and engaging manuscript. Authors present a systematic study of how humans perceive chasing from low level motion characteristics. The rationale for each step is appropriate and the methods chosen align with the research questions.

I have a few queries, although I see these queries as something that the authors may wish to consider to enhance the paper – not as fundamental flaws.

1. Why are both frequentist and Bayesian statistics reported? Frequentist and Bayesian statistics have fundamentally different philosophies that underlie them, and it is a little strange to double up. It is probably best to choose the statistical philosophy that aligns better with your own philosophy or is most appropriate for the questions that you are asking.

We appreciate the reviewer’s comment and perspective. The reason why we chose to use both frequentist and Bayesian statistics is rooted in recent arguments that psychologists should provide more information that would allow readers to evaluate null results (e.g., https://doi.org/10.1177/251524591877374). Given this argument, our primary goal was, when there was a null result reported, to provide somewhat easily interpretable information about the strength of the null (which frequentist statistics are not really meant for). We ended up providing those statistics in all cases, even when frequentist statistics were not null, out of concern that readers would find it difficult to navigate when BFs were only provided for some analyses. 

However, the reviewer’s comment makes us more comfortable in removing Bayesian statistics in cases where frequentists results were significant (and to those we added measures of effect size as suggested). For null results, we kept both frequentist and Bayesian statistics to ensure readers can interpret the strength of the null results. We also added a paragraph to explain our rationale for using both types of statistics (line 311-317).

2. If you stick with frequentist statistics, I advise reporting effect sizes for all tests (there are many without). Sometimes it appears as though the BF is taking the place of the effect size because the BF factor appears where the effect size should go. To avoid confusion, it probably should be separated clearly so readers do not assume that the authors are trying to use the BF factor as the effect size of a frequentist calculation.

As suggested, we have added effect sizes for all frequentist statistics.

3. I had difficulty understanding the correlation-control wolf. It might be worth highlighting that the videos are available on the OSF and making example videos for the ones that are in matlab code (not everyone will have access to matlab given that it is not free).

We apologize that the explanation of the correlation-control wolf was not clear. As suggested, we have highlighted that the videos are available on OSF (line 149) and we added a sample video of the correlation-control wolf.

4. I am note sure if ecologically It would be more important to identify a chasing agent (pg. 23). In many cases the opposite might be true. Consider hunting in a pack, which our evolutionary ancestors would have done (and one of our closest evolutionary ancestors do – chimpanzees), perhaps it is more important to identify the relationship between the chaser and chased? If one of your hunting group is focused on a particular individual prey you would be best to focus your efforts on that prey – then it becomes important to identify the relationship between the chaser and chased – without knowledge of both it will make it difficult to succeed (and communicative cues would alert the chased more). In other words, it is very important to know which sheep is being chased, otherwise group chasing strategy would be poor.

Thank you for bringing up this possibility. It’s not a possibility we had initially considered, but it’s an excellent point. We have therefore added this to our manuscript (line 606-610). 

I sign all my reviews regardless of their content; be it a positive or negative evaluation. I aim to provide high quality, thorough, considerate and respectful reviews. Transparency assists in meeting those aims. Should this review not meet those aims I am always happy to take feedback on my review.

Sincerely,

Merryn Constable

Reviewer #2: Summary

Past research on the visual detection of chasing movements has found that direct (heatseeking) chasing is more detectable. The current research demonstrates that when two participants engage in "real" chasing, their movement is not nearly as heatseeking as the algorithm that has typically been studied. It also confirms in various ways the past finding that heatseeking chasing is more detectable than other types, while also finding some sensitivity to other kinds of chasing.

In Experiment 1, two subjects played a game against one another, in which they used the arrow keys to control dots, and one subject chased the other. The main contribution of the paper was to discover that, in contrast to the simple heatseeking chasing algorithm used in previous research, subjects did not chase directly towards each other, but rather followed a pattern of indirect subtle chasing. In part 2 of this experiment, a new group of subjects was asked to perform & identify the chasers; task involving either algorithmically heatseeking trajectories or recordings of the realistic chasing from part 1. Consistent with past research showing that chasing is less detectable when it is more indirect, subjects were more sensitive to heatseeking chasing than the more indirect realistic chasing.

In Experiment 2, accuracy identifying the wolf was again measured, comparing three algorithms — heatseeking, predictive chasing, and displays in which the wolf copied the sheep. The important discovery in this experiment was that performance was again best for heatseeking chasing, even though predictive chasing was arguably more efficient and more naturalistic.

In Experiment 3, the same chasing algorithms were varied, and subjects identified either the wolf or the sheep, and were better at detecting the sheep, probably because it moved in a more distinctive way by circling the perimeter of the display. (Also, one of the distractors chased the sheep's invisible location, and subjects were not likely to falsely identify this distractor as chasing. This is consistent with past claims that merely correlated movement of a dot chasing the invisible mirror image of another dot in the display does not look like chasing.)

In Experiment 4, subjects again failed to detect a wolf when it chased an invisible sheep.

Overall, this article was an interesting read, but could benefit from some revisions.

Big Points

In Experiment 2, comparing sensitivity to heatseeking vs. more naturalistic and effective "interception" chasing seems well motivated. But it is less clear what is learned from testing sensitivity to "mimicking" chasing. It seems unlikely that subjects in the task actually spontaneously saw this as chasing (maybe this is the authors’ discovery, but why is that important?). So we would recommend rerunning Experiment 2 without that condition — which would also allow an opportunity to rerun that experiment without including an awkwardly unanalyzed task. Or perhaps the authors could better explain the purpose of including this condition.

We appreciate the reviewer’s comment as it made it clear we should have provided more text outlining the choice of a mimicking agent. In short, while a mimicking approach may not be effective as a method to catch a target in an unbounded environment (e.g., a wide-open plain), it has reasonable utility in the context of a bounded environment, which is obviously the type of environment that our research, like all previous work has considered. Notably this type of bounded environment does have potential analogs in the real-world – both due to man-made constructions (e.g., chasing within buildings, within fences, in dense cities, etc.), but also in the natural world (e.g., where environmental features like rivers, lakes, mountains, cliffs, etc. create boundaries). We have now added text to this end in the manuscript (line 386-397). 

We would also recommend removing Experiment 3, in which there was better detection of the sheep than the wolf, since the reason for this is almost certainly just a confound in the sheep movement looking more distinctive.

While we certainly appreciate the reviewer’s point, we would respectfully disagree that Experiment 3 provides so little value to the field that it should be removed entirely. First, no previous experiment has tested whether performance is equivalent between detecting the chasing agent and the agent being chased. Second, we’d note that the circling algorithm was not a strawman, but was chosen based on pilot data indicating that this was a common strategy employed by human participants playing the role of sheep. We would therefore argue that there is some utility in knowing whether there are asymmetries in the ability to detect the chaser and chasee. Finally, even though the sheep in our design only used one movement algorithm, our results indicated that while performance was numerically similar across the three conditions (heat-seeking, mimicking, and interception) they were not necessarily identical. Instead, it appears that the sheep was numerically most difficult to detect in the context of a wolf that was utilizing a trajectory intercept strategy. This would make sense from base principles if the perception of the chaser/chase relationship is inherently about a dyad (i.e., as suggested in the discussion section, if things like proximity and correlated motion cues matter for detecting the chaser, then if those are weaker, it *should* also impact the ability to perceive the chasee). We have thus elaborated on this point in the manuscript (line 788-804). 

Small Points

Looking at the videos, it seems that the distractors on average moved within a small area, whereas the wolf and sheep covered much longer distances, and were more peripheral. Across studies, could this have played a role in the patterns of results you observed?

We agree with the reviewer that it is indeed the case that some distractors moved within a small area. However, based on our algorithm, other distractors would also cover even longer distances than the wolf and the sheep. Thus, we do not believe that aspect of the distractors’ movement pattern would have played a role in our results. However, as in all studies of this type, the problem for the participant is inherently one about understanding contrasts (i.e., what makes something a wolf in the context of a certain set of distractors). We attempted to make our background distractors as uninformative as possible, but we agree this is useful to make explicit and have done so in the revised manuscript. 

Lines 56-65: Up to the authors, but it’s possible that this section can be made more concise.

In our experience, this has often been a useful way to introduce individuals that are less well-versed in the specific field, and so our preference would be to keep the text as is. We would of course be happy to make changes if determined appropriate by the editor. 

Line 129: Maybe change "two additional issues" to "two additional issues, which have been explored in past research", since past work on chasing has already studied both the effects of # of distractors and whether correlated movement of this sort is sufficient to see chasing.

We revised our manuscript based on the reviewer’s suggestion (line 130).

Line 262: Not ideal that the experimenters had a hand in selecting/trimming the naturalistic chasing stimuli that got used for part 2. Without this curation, is it possible that the naturalistic chasing stimuli would have been detected as well as the heatseeking chasing stimuli in part 2?

Thank you for pointing this out. There were several reasons why we manually selected/truncated the videos. First, in terms of selecting/removing videos - we removed videos from the stimulus set if they were shorter than 6 seconds. In pilot work, videos less than this length were generally too short for participants to detect any chasing behaviors. Then, with respect to truncating, in cases where the sheep was caught by the wolf, this truncation was somewhat by necessity if the videos were to be included at all (i.e., it was done to include as many trials as possible). If, in the video, the wolf eventually caught the sheep, the task would become somewhat trivial, as the disc closest to the sheep would be the wolf. The truncated portion of the videos were generally no longer than 2 seconds. Finally, for trials that lasted the entire 30 seconds and were thus cut to 15 seconds (i.e., trials which the sheep successfully escaped the wolf), we did not see any significant evidence that the second half of the trial proceeded very differently than the first and thus our preference was to ensure that the distribution of starting sheep positions was somewhat equivalent in the set (which would have been harder to do if truncating in an alternate way). 

We agree with the reviewer though that this could be an important point for researchers hoping to replicate or extend our work and thus we have added explanations in the manuscript to better justify our manual selection/trimming (line 265-266 and line 273-276).

Line 279: What was the stopping rule for data collection?

In practice, our stopping rule (after we had exceeded our minimum hoped for number of participants) was related to participant availability (i.e., given the lack of strong quantitative a priori data to base predictions on, our disposition was to collect as many participants as possible within our time window). 

Line 334: It would be good to include more explanation of why these three particular algorithms were tested. The "copying" condition in particular seems unmotivated.

As per the comment above, we have added text to further motivate this condition (line 788-804). 

Line 434: Actively controlling the sheep to escape the wolf has been used as a measure of the perception of intentionality in the past, so not all readers will be convinced that this task as not relevant.

We thank the reviewer for this valuable suggestion and agree the active controlling task is relevant to participants’ ability to perceive chasing. However, in the current manuscript, we only consider participants’ ability to perceive chasing behaviors in passive observation tasks. Thus, we consider the active controlling task to be not relevant to our purpose of study in the current manuscript. In our research program (which makes up the first author’s dissertation work), the active controlling task was motivated by a separate set of research questions and thus is planned to be presented in a separate manuscript. In addition, as we agree that the active controlling task is relevant to participants’ overall ability to detect chasing behaviors, we removed passive observation trials that were done after the participants completed the active controlling task (the latter 48 trials mentioned on line 436 in the original manuscript) to avoid any misinterpretation. 

Line 464: Subjects in the wolf-mimics-sheep condition may have learned to respond to this stimulus, but just because they succeeded (albeit less so than when there was heatseeking or predictive chasing), does this mean that they really saw it as an intentional behavior? (Or is the claim that their failure to detect this was due to not seeing it as intentional?) How it looked to subjects seems like an important point, if one of the main claims of the article (e.g. on line 124) is that subjects’ ability to detect different kinds of chasing above chance implies that they are perceiving the underlying cause/goal of the chasing rather than something more superficial about chasing movements.

The reviewer’s point regarding whether it’s possible to simultaneously succeed at the task (i.e., identify the wolf), but nonetheless not see the behavior as intentional, is an interesting one. To some extent, it’s not clear that our data for any of the wolf-types – not just the mimicking one – could falsify that hypothesis as we did not ask any more subjective or qualitative questions about the participant’s perceptions (and actually, that’s true of a great deal of the work to-date). 

In our case we started with the view that all of our wolf-types *are* intentional agents – they have a clearly specified goal and are executing behaviors in the service of that goal (and all of them do a reasonably good job of meeting that goal). The question then is whether all types of the chasing (or “catching”) intention are equivalently easy for human participants to detect/identify. The fact that this wasn’t found to be uniformly true provides some nuance to the story – in that it suggests that there is, at a minimum, some sort of hierarchy of templates that humans uses when they are looking for an agent showing the “chasing intention.” 

Line 615-621: Haven’t wolves chasing an invisible sheep commonly been used in past chasing detection experiments as a target-absent display? It might be better to say this outright, and to explain how Experiment 4 teaches us something new.

We appreciate this suggestion and have revised the manuscript accordingly to better indicate what new information is gleaned from Experiment 4 (line 700-704). 

Line 719: Arguably this could explain the constant performance across the three conditions (e.g. if the sheep looked very distinctive in all three conditions on the basis of its very different movement, leading it to be quite detectable regardless of the wolf’s style of chasing).

We agree it is possible that the circling sheep made more distinctive and thus were easy to identify regardless of what strategy was used by the wolf, however, if the sheep’s movement pattern was distinctive enough, we would expect to see participants’ performance at ceiling. However, participants’ performance in the sheep identification task was far from ceiling (M = 72%). Furthermore, the statistics indicate a main effect of wolf-type, but no interaction with task. We’re thus wary to interpret the results of the sheep identification as identical (as we note above, if performance was identical across wolf-types while still being well below ceiling, it would speak deeply to theories regarding the importance of shared information across the dyads; however, we’re not sure our data is so strong as to make a substantial point in this direction). 

We would like to thank the reviewers for evaluating our manuscript. We have tried to address all the reviewers’ concerns in a proper way and believe that our paper has improved considerably. We would be happy to make further corrections if necessary and look forward to hearing from you soon.

---

## [Decision Letter · Decision Letter 1]

27 Mar 2023

PONE-D-22-29587R1Realistic and complex visual chasing behaviors trigger the perception of intentionalityPLOS ONE

Dear Dr. Ji,

Thank you for submitting your manuscript to PLOS ONE. After careful consideration, we feel that it has merit but does not fully meet PLOS ONE’s publication criteria as it currently stands. Therefore, we invite you to submit a revised version of the manuscript that addresses the points raised during the review process.

While both reviewers are generally satisfied with your revisions, Reviewer 1 points out some potential confusion about statistical reporting. Please address and clarify this point carefully. 

We look forward to receiving your revised manuscript.

Kind regards,

Guido Maiello

Academic Editor

PLOS ONE

Journal Requirements:

Reviewers' comments:

Reviewer's Responses to Questions

**Comments to the Author**

1. If the authors have adequately addressed your comments raised in a previous round of review and you feel that this manuscript is now acceptable for publication, you may indicate that here to bypass the “Comments to the Author” section, enter your conflict of interest statement in the “Confidential to Editor” section, and submit your "Accept" recommendation.

Reviewer #1: (No Response)

Reviewer #2: All comments have been addressed

2. Is the manuscript technically sound, and do the data support the conclusions?

Reviewer #1: Partly

Reviewer #2: Yes

3. Has the statistical analysis been performed appropriately and rigorously? 

Reviewer #1: No

Reviewer #2: Yes

4. Have the authors made all data underlying the findings in their manuscript fully available?

Reviewer #1: Yes

Reviewer #2: Yes

5. Is the manuscript presented in an intelligible fashion and written in standard English?

Reviewer #1: Yes

Reviewer #2: Yes

6. Review Comments to the Author

Reviewer #1: I am not sure if the authors uploaded an earlier manuscript given that they have said that effect sizes have now been provided. They have not been provided and the Bayes factors still appear as though they are being used as an effect size for the frequentist statistics. Given that frequentist and Bayesian philosophies are fundamentally different and based on different assumptions this is inappropriate.

An appropriate way of following up null results in frequentist statistics is equivalence testing:

Lakens, D., Scheel, A. M., & Isager, P. M. (2018). Equivalence Testing for Psychological Research: A Tutorial. Advances in Methods and Practices in Psychological Science, 1(2), 259–269. https://doi.org/10.1177/2515245918770963

Where I can accept that the authors may wish to provide both Frequentist and Bayesian statistics they should be clearly separated so one is not used as an effect size for the other. I believe that potentially my review was misunderstood.

Here is a primer on effect sizes for frequentist statistics:

Lakens, D. (2013). Calculating and reporting effect sizes to facilitate cumulative science: A practical primer for t-tests and ANOVAs. Frontiers in Psychology, 4. https://doi.org/10.3389/fpsyg.2013.00863

Sincerely,

Merryn Constable

Reviewer #2: The authors have responded to my comments. I find their responses to be principled, and have no further comments to add.

7. PLOS authors have the option to publish the peer review history of their article (what does this mean?). If published, this will include your full peer review and any attached files.

Reviewer #1: No

Reviewer #2: No

---

## [Author Response · Author response to Decision Letter 1]

29 Mar 2023

Response to Reviewers

We would like to again thank the reviewer’s time and effort to comment on the manuscript and edited the manuscript to address their concerns. Below you can find responses to each concern raised by the reviewers. 

Reviewer #1: I am not sure if the authors uploaded an earlier manuscript given that they have said that effect sizes have now been provided. They have not been provided and the Bayes factors still appear as though they are being used as an effect size for the frequentist statistics. Given that frequentist and Bayesian philosophies are fundamentally different and based on different assumptions this is inappropriate.

An appropriate way of following up null results in frequentist statistics is equivalence testing:

Lakens, D., Scheel, A. M., & Isager, P. M. (2018). Equivalence Testing for Psychological Research: A Tutorial. Advances in Methods and Practices in Psychological Science, 1(2), 259–269. https://doi.org/10.1177/2515245918770963

Where I can accept that the authors may wish to provide both Frequentist and Bayesian statistics they should be clearly separated so one is not used as an effect size for the other. I believe that potentially my review was misunderstood.

Here is a primer on effect sizes for frequentist statistics:

Lakens, D. (2013). Calculating and reporting effect sizes to facilitate cumulative science: A practical primer for t-tests and ANOVAs. Frontiers in Psychology, 4. https://doi.org/10.3389/fpsyg.2013.00863

We apologize for the misunderstanding. For our revision, we uploaded a new manuscript version corresponding to the changes described in our revision letter, but did not delete the original manuscript (given the way thet submission portal is set up, it wasn’t clear how the system wanted us to do version control when building the pdf for reviewers). We have now removed the previous submissions and only kept the most recent revised manuscript (both with track changes and without). We agree it is inappropriate to treat Bayes’ Factors as an effect size for frequentist statistics and these are clearly indicated in the revised manuscript. 

We would like to thank the reviewers for evaluating our manuscript. We have tried to address all the reviewers’ concerns in a proper way and believe that our paper has improved considerably. We would be happy to make further corrections if necessary and look forward to hearing from you soon.

---

## [Decision Letter · Decision Letter 2]

3 Apr 2023

Realistic and complex visual chasing behaviors trigger the perception of intentionality

PONE-D-22-29587R2

Dear Dr. Ji,

We’re pleased to inform you that your manuscript has been judged scientifically suitable for publication and will be formally accepted for publication once it meets all outstanding technical requirements.

Kind regards,

Guido Maiello

Academic Editor

PLOS ONE

Additional Editor Comments (optional):

Reviewers' comments:

Reviewer's Responses to Questions

**Comments to the Author**

1. If the authors have adequately addressed your comments raised in a previous round of review and you feel that this manuscript is now acceptable for publication, you may indicate that here to bypass the “Comments to the Author” section, enter your conflict of interest statement in the “Confidential to Editor” section, and submit your "Accept" recommendation.

Reviewer #1: All comments have been addressed

2. Is the manuscript technically sound, and do the data support the conclusions?

Reviewer #1: Yes

3. Has the statistical analysis been performed appropriately and rigorously? 

Reviewer #1: Yes

4. Have the authors made all data underlying the findings in their manuscript fully available?

Reviewer #1: Yes

5. Is the manuscript presented in an intelligible fashion and written in standard English?

Reviewer #1: Yes

6. Review Comments to the Author

Reviewer #1: (No Response)

7. PLOS authors have the option to publish the peer review history of their article (what does this mean?). If published, this will include your full peer review and any attached files.

Reviewer #1: No

---

## [Editor Report · Acceptance letter]

5 Apr 2023

PONE-D-22-29587R2 

Realistic and complex visual chasing behaviors trigger the perception of intentionality 

Dear Dr. Ji:

I'm pleased to inform you that your manuscript has been deemed suitable for publication in PLOS ONE. Congratulations! Your manuscript is now with our production department. 

Kind regards, 

on behalf of

Dr. Guido Maiello 

Academic Editor

PLOS ONE